# Linking LOXL2 to Cardiac Interstitial Fibrosis

**DOI:** 10.3390/ijms21165913

**Published:** 2020-08-18

**Authors:** Melisse Erasmus, Ebrahim Samodien, Sandrine Lecour, Martin Cour, Oscar Lorenzo, Phiwayinkosi Dludla, Carmen Pheiffer, Rabia Johnson

**Affiliations:** 1Biomedical Research and Innovation Platform, South African Medical Research Council, Cape Town 7501, South Africa; melisse.scheepers@mrc.ac.za (M.E.); ebrahim.samodien@mrc.ac.za (E.S.); phiwayinkosi.dludla@mrc.ac.za (P.D.); carmen.pheiffer@mrc.ac.za (C.P.); 2Department of Medical Physiology, Stellenbosch University, Cape Town 7505, South Africa; 3Hatter Institute for Cardiovascular Research in Africa (HICRA), University of Cape Town, Cape Town 7925, South Africa; sandrine.lecour@uct.ac.za; 4Hospices Civils de Lyon, Hôpital Edouard Herriot, Service de Médecine Intensive-Réanimation, Place d’Arsonval, 69437 Lyon, France; martin.cour@chu-lyon.fr; 5Institute de Investigación Sanitaria-FJD, Faculty of Medicine, University Autónoma de Madrid, 28049 Madrid, Spain; olorenzo@fjd.es; 6Spanish Biomedical Research Centre in Diabetes and Associated Metabolic Disorders (CIBERDEM) Network, 28040 Madrid, Spain

**Keywords:** Lysyl Oxidase-Like 2 (LOXL2), epigenetics, DNA methylation, fibrosis, cardiovascular disease (CVD)

## Abstract

Cardiovascular diseases (CVDs) are the leading causes of death worldwide. CVD pathophysiology is often characterized by increased stiffening of the heart muscle due to fibrosis, thus resulting in diminished cardiac function. Fibrosis can be caused by increased oxidative stress and inflammation, which is strongly linked to lifestyle and environmental factors such as diet, smoking, hyperglycemia, and hypertension. These factors can affect gene expression through epigenetic modifications. Lysyl oxidase like 2 (LOXL2) is responsible for collagen and elastin cross-linking in the heart, and its dysregulation has been pathologically associated with increased fibrosis. Additionally, studies have shown that, LOXL2 expression can be regulated by DNA methylation and histone modification. However, there is a paucity of data on LOXL2 regulation and its role in CVD. As such, this review aims to gain insight into the mechanisms by which LOXL2 is regulated in physiological conditions, as well as determine the downstream effectors responsible for CVD development.

## 1. Introduction

According to the latest World Health Organization (WHO) report, cardiovascular diseases (CVDs) account for approximately 31% of all global deaths [1]. To curb this high incidence of CVD, a new initiative from the WHO in collaboration with the United Nations was launched with the sole purpose of up-scaling efforts to prevent and control this global threat of cardiovascular deaths, and focuses on nutritional interventions to reduce CVD risk factors such as obesity, diabetes and hypertension [2,3]. An estimated 463 million individuals were reported to have diabetes in 2016, which can partially be attributed to the spread of the Western diet and decreased physical demand of the modern-day [4]. According to the latest IDF report, the observed increased diabetes prevalence and its associated cardiovascular risk is driven by a complex interplay of socioeconomic, demographic, environmental and genetic factors [5]. Furthermore, diabetes is a known risk factor for cardiovascular dysfunction, with approximately 85% of CVD deaths occurring as a result of heart failure (HF) or stroke [6]. There are certain lifestyle choices, such as smoking, physical inactivity and consumption of foods high in fat, sugar and salt that contribute, to the observed increase in metabolic diseases and subsequent cardiovascular risk [7]. Furthermore, it has been argued that this increased cardiovascular risk is associated with chronic low-grade inflammation which is known to promote cardiac fibrosis and stiffening of the heart muscle [8]. Myocardial stiffening results in HF, which is caused by an inability of the heart to pump blood through the body effectively, thereby resulting in decreased myocardial compliance which has been attributed to alterations in cross-linked collagen [9].

A healthy heart is composed of cardiomyocytes enveloped in cross-linked collagen, the main component making up the extracellular matrix (ECM), which is critical for maintaining the structural integrity of the myocardium [10]. In conditions of cardiac damage, such as those induced by obesity or diabetes, an abnormal increase in collagen deposition is observed between the myocytes, which leads to the thickening of the ventricle wall, thus resulting in the stiffening of the heart muscle [11]. Moreover, dysregulation of cardiac tissue structure that results in increased ECM protein deposition is known to cause myocardial stiffness and dysfunction, further raising the risk for HF [12,13]. Kasner and colleagues (2011) showed that patients with diastolic dysfunction, and those with HF with normal ejection fraction exhibited an upregulation in collagen type I expression that was accompanied by increased collagen deposition and fibrosis [14]. These findings suggest that optimal regulation of the ECM and controlling collagen deposition is critical in maintaining normal cardiac function. The mechanism of action of how ECM dysregulation contributes to fibrotic disease and CVD is not fully elucidated; thus, a better understanding of this process would add value to current knowledge of drivers of CVD and cardiac fibrosis.

Collagen is the most abundant protein in the ECM, and in the human body, types I-III are the most prominent [15]. Type I collagen is a fibrillar collagen known to be associated with an increase in fibrosis [16]. It consists of two α1 and one α2 polypeptide chains, unlike type II and III collagens, which consist of three α1 peptide chains [17]. Once procollagen has been secreted by the fibroblasts [18], the propeptide regions are then truncated to form tropocollagen [19]. The telopeptide regions of the collagen molecules contain lysine and hydroxylysine residues which are responsible for crosslink formation through chemical bonds, once the collagen molecules have self-assembled [19]. Lysyl oxidase (LOX) aids in the crosslinking by oxidative deamination of the lysine and hydroxylysine residues, which then allow them to form the crosslinks [19].

LOX and its linked LOX-like (LOXL) isoforms are known to play a major role in ECM remodeling, and their physiological regulation has become an important mechanism implicated in the development of connective tissue disorders, including cardiac tissue fibrosis. Certainly, evidence has been summarized on the molecular biology of the LOX family proteins in relation to its role in tumor biology [20,21]. Most recently, Rodríguez and Martínez-González [22,23] discussed findings on the role of the LOX family in the pathogenesis and progression of myocardial disorders. Similarly, Yang (2016) and later Zhao (2017) reported on the role that lysyl oxidase-like 2 (LOX2) plays in the development of cardiac fibrosis and subsequent heart failure [24,25]. More importantly, although such information remains of interest, specialized information on the regulation of LOXL2 in relation to the pathophysiology of cardiac fibrosis remains scarce. This is of interest since LOXL2 is a protein that plays an essential role in the biogenesis of collagen deposition and cross-linking and is thus increasingly being explored for its role in the development of cardiac fibrosis [24].

It is a copper-dependent enzyme that catalyzes the cross-linking of collagen fibers and thereby regulates collagen homeostasis, consequently conferring ECM stability [24,26]. Basal levels of LOXL2 expression are necessary for normal ECM deposition and structural composition in tissue and is central to the functionality and maintenance of the myocardium [27]. Miner et al. (2006) argued that dysregulation through an augmented expression of LOXL2, increased collagen deposition between myofibroblasts and is a major driver in the development of cardiac fibrosis, causing heart muscle stiffness and reduced cardiac output [11]. Fong et al. (2007) confirmed this and argued that decreasing excessive collagen cross-linking would potentially reduce myocardial stiffness and improve heart function [28]. They reported that LOXL2 expression may be regulated by DNA methylation and histone modification [28]. As such, it suggests that understanding disease pathophysiology and epigenetic changes that are causal drivers of CVD pathology may allow for insight into the mechanistic action of LOXL2 and its potential use as a marker of cardiovascular dysfunction.

As such, the current review will describe the function of LOXL2 in maintaining ECM homeostasis, with the aim of understanding the involvement of LOXL2 in the pathophysiology of cardiac fibrosis. Additionally, the role that epigenetic mechanisms play in the regulation of LOXL2 expression will also be explored.

## 2. The ECM is Compromised during the Development of Fibrosis

In the heart, the ECM forms a scaffold, supporting the myocardial cells and blood vessels and plays a crucial role in cellular structure and repair, as well as angiogenesis [10,11]. When the heart muscle is relaxed, the ventricles expand, allowing blood to enter the ventricular space [29]. During contraction, however, an electric action potential is generated in the sinoatrial node of the right atrium, which in turn, causes a release of calcium from the sarcoplasmic reticulum [30,31]. This increased calcium in the cytosol binds to troponin which triggers muscle contraction by allowing the myosin heads to bind to the actin binding sites. The subsequent conformational change in the myosin results in the contraction of the ventricles and forces blood out of the heart to be pumped throughout the body [32]. Collagen fibers within the ECM store the energy generated during the contraction of the heart and aid in myocardial relaxation, where cardiomyocytes lengthen and return to their original state [33].

Collagen is the most abundant protein in the body and the main component of the ECM; it plays a key role in the maintenance of cardiac tissues strength and structural integrity [34,35,36]. Indeed, collagen deposition is key to tissue remodeling and wound healing [37]; however, excessive accumulation of collagen and other extracellular matrix components in the heart can be detrimental, leading to scarring and fibrosis of cardiac tissue [35,37,38,39]. Mechanistically it has been speculated that the abnormal accumulation of the collagen diminishes the myocytes’ contractility in the affected area, while the overall heart compliance and diastolic function are reduced. For example, in a study done by Querejeta et al. (2004) it was argued that an increase in collagen type I in the heart resulted in an increase in myocardial fibrosis, which was observed in hypertensive patients with left ventricular hypertrophy [40]. Similarly, in a study done by Martos et al. (2007), it was reported that ventricular stiffness due to increased collagen deposition resulted in diastolic heart failure or severe diastolic dysfunction in hypertensive patients [41].

The latter studies confirm that apart from its role in maintaining the structural integrity and stability of the myocardium [10], the ECM, and so too, enhanced collagen cross-linking plays a major role in the scaring and fibrotic processes which lead to HF [11]. In the instance of excessive collagen and ECM protein deposition between the myofibroblasts, the resultant thickening of the ventricle wall has been shown to cause confinement of the muscle fibers, restricting their movement past each other [11]. This collagen build-up eventually promotes fibrosis causing stiffening of the heart muscle, resulting in suboptimal contractile and pump function [11]. Therefore, a better understanding of the molecular mechanisms that drive this process, specifically collagen homeostasis, may lead to the development of possible novel antifibrotic therapies. According to Horn (2016), increased collagen cross linking may occur via two mechanism: (i) through augmented advanced glycation end product (AGE) formation, and (ii) enhanced lysyl oxidase (LOX)-mediated aldehyde formation [42].

LOXL2 is a key enzyme that catalyzes the cross-linking of collagen fibers and is integral for collagen homeostasis. It is thus needed for normal functioning of the myocardium and is pertinent to cardiac remodeling [13]. Dysregulation of its expression is a major driver of muscle stiffness through induced cardiac fibrosis [24,25,43], which reduces cardiac output. In fact, it has been proposed that decreasing excessive collagen cross-linking would reduce myocardial fibrosis and stiffness and thereby improve heart function [44]. Thus, finding methods for controlling collagen deposition and cross-linking through LOXL2 could be a monumental breakthrough in protecting heart against fibrosis, and eventually reducing the burden of CVDs.

## 3. The Lysyl Oxidase Gene Family

Lysyl oxidase (LOX) and lysyl oxidase-like 1–4 (LOXL1–4) are a family of proteins that play an essential role in collagen and elastin cross-linking [45]. It is now well established that the LOX protein family is linked to fibrosis, as well as the development of other connective tissue disorders, such as elastolysis and Ehlers-Danlos syndrome [46]. Although differentiation between the various isoforms is still unclear, LOX is known to promote collagen and elastin cross-linking by oxidatively deaminating the lysine and hydroxylysine groups within the peptide chains [46]. In addition, LOX is implicated in the transcriptional regulation of transforming growth factor beta (TGF-β) signaling [47]. Increased TGF-β is known to upregulate of alpha smooth muscle actin (α-SMA) leading to excessive cardiac scar formation with enhanced fibrosis [36,48].

TGF-β and α-SMA form an integral part in the fibrotic signaling pathways along with LOX. Although LOX and the LOXL1–4 isoforms are suspected to function similarly due to the conservation of the catalytic domains, the LOX protein family is predominantly expressed in different tissue types with different substrate preferences [26]. It is known that some of the tissue and substrate targets of the different LOX isoforms may overlap, but the exact localizations and functions of these proteins are not yet clearly defined. As illustrated in Figure 1, the copper binding, lysine tyrosylquinone cofactor, cytokine receptor-like and C-terminal regions are catalytic domains that are conserved across the LOX protein family [49,50,51]. These conserved domains are necessary for LOX activity, and due to these conserved domains and high protein similarity, it is predicted that LOXL 1–4 isoforms perform cross-linking in a similar manner. Any variations in the function of the LOX-family isoforms is likely to occur due to differences in the variable N-terminal regions of each protein.

LOX exists in an inactive form, which is mainly located in the cytosol, and upon activation, it is translocated to the nucleus and endoplasmic reticulum, and secreted into the extracellular space [52]. The inactive form is associated with the endoplasmic reticulum and receives the copper ion cofactor from Menkes’ protein, a copper transporter, to produce the active pro-LOX [26]. Once in the extracellular space, the activated LOX is known to interact with procollagen molecules, and oxidatively deaminates lysine and hydroxylysine residues in the telopeptide portions of the fibers to form lysine aldehyde and hydroxylysine aldehyde, respectively [19]. Once the propeptide fragments of the procollagen molecules are truncated, forming tropocollagen molecules, they self-assemble, and the aldehyde portions form covalent bonds to cross-link the collagen molecules to form fibrils [19]. The mechanisms involved in LOX-dependent cross-linking are depicted in Figure 2, with the process thought to be similar for LOXL1-4.

Apart from collagen and elastin cross-linking, the LOX protein isoforms perform additional functions because of the substrate preference of each enzyme. For example, LOX knock-down studies have shown that it is necessary for the proliferation and differentiation of some cell types, such as osteoblasts [53]. LOX also has the ability to inactivate TGF-β and fibroblast growth factor 2 (FGF-2) by means of oxidation, thus controlling the differentiation of myofibroblasts to fibroblasts [54,55]. Liu et al. (2004) reported that LOXL1 displays substrate preference for elastin and in knockout mouse studies, has been associated with development of abnormal vasculature [56]. In addition, it was shown by Ohmura et al. (2012) that transgenic expression of LOXL1 in mouse hearts induced cardiac hypertrophy [57]. Like LOXL1, Busnadiego et al. (2013) [58] reported that the LOXL4 isoform is involved in vascular remodeling, although it has a preference for collagen as a substrate. Similarly, LOXL3 has been implicated in vascular remodeling and the development of cartilage and was found to be highly expressed in mesenchymal cells, with an increased substrate affinity for collagen type XI alpha 1 and 2 (COL11A1, COL11A2) and collagen type II alpha 1 (COL2A1), the main structural component of cartilage [59]. A study by Zhang et al. (2015) showed that homozygous LOXL3 knockout mice resulted in perinatal fetal death, and LOXL3 +/− mice were found to have spinal abnormalities and cleft palate [60]. As a result, LOXL3 knockout animals could not be used for studying LOXL3 effect on cardiovascular development or changes. Lastly, LOXL2 uses collagen type IV as its preferred substrate and cross-links it through both enzymatic interactions, by the active protein, and non-enzymatic interactions, by the inactive form of the protein [61]. It is proposed to play a role in a variety of functions such as normal bone development, blood vessel stabilization, and the sprouting of new blood vessels, with LOXL2 often found localized within endothelial cells [61]. In addition, as with LOXL3, studies using LOXL2 knockout mice observed perinatal fetal death as a result of hepatic and cardiovascular defects [62]. This was confirmed by Yang et al. (2016), who demonstrated that LOXL2 secretion increased in stressed mouse hearts, triggering fibrosis [24]. Taken together, the mentioned studies implicated LOXL2 in myocardial function; however, research pertaining to LOXL2 in the heart is lacking, and thus its role in cardiac fibrosis and disease will be further discussed. Given the crucial function of LOXL2, further studies are required to understand what factors govern its regulation and function in cardiac health and disease.

## 4. LOXL2 in Disease

Although the *LOXL2* gene has been most widely studied in cancer [50], it has recently attracted interest for its role in fibrotic diseases such as idiopathic pulmonary fibrosis and liver fibrosis during non-alcoholic steatohepatitis [63,64]. As with the previous findings by Yang and Zhao, in a study by Johnson et al. (2020), the results confirm the involvement of LOXL2 in cardiac dysfunction [65]. Nonetheless, evidence on the physiological regulation and data pertaining to the role of LOXL2 in cardiac health has not been reviewed. As such, understanding the mechanism by which LOXL2 is implicated in fibrosis during the pathogenesis of various medical conditions, like CVDs, may pave the way for future therapeutic interventions.

In a healthy state, fibroblasts are present in tissue in an inactive state. After injury, the fibroblasts are activated and differentiate into myofibroblasts in the presence of growth factors and cytokines, such as TGF-β, which stimulate ECM proteins to aid in the healing response [38]. It is known that under conditions of stress, such as inflammation, LOXL2 expression is induced, and its secretion into the extracellular space is increased [66]. Lytle et al. (2017) found, in low-density lipoprotein receptor deficient mice, that consumption of a high-fat, high-sugar diet increased plasma *LOXL2* mRNA levels [67]. This was confirmed by Yang et al. (2016), who showed that augmented LOXL2 expression causes an increase in TGF-β signaling through activation of the phosphoinositide 3-kinase/protein kinase B/mechanistic target of rapamycin (PI3K/AKT/mTOR) pathway [24] (Figure 3). Furthermore, they showed that TGF-β induced myofibroblasts formation, subsequently increasing alpha smooth muscle actin (α-SMA) [24], which was previously implicated in increased collagen deposition during the fibrotic response and scar tissue formation [38,64]. In this regard, Trackman (2016) concluded that LOXL2 not only plays a role in the collagen cross-linking, but can also activate alternative fibrotic pathways through the recruitment of fibroblasts [26]. Since LOXL2 is implicated in fibrosis, it is crucial to understand its role in the pathophysiology of fibrosis-induced cardiac dysfunction.

### The Role of LOXL2 in the Development of Cardiovascular Disease

Diabetes is a risk factor for cardiac dysfunction and often occurs as a result of increased oxidative stress in the cells [68]. The combination of oxidative stress and the presence of high amounts of glucose in the blood leads to the production of AGEs, which can accumulate in various organs including the kidneys and heart, leading to vascular diseases and endothelial dysfunction [69,70,71]. AGEs are known to aggregate on proteins involved in fibrotic processes, such as fibronectin and collagen, and can alter the normal degradation of the proteins [72]. When AGEs link with collagen molecules, it has been found to decrease elasticity and increase stiffness within the vasculature and the heart tissue [73]. This leads to myocardial fibrosis by the induction of transcription factor binding, which promotes the expression of genes such as LOXL2 [73].

Cardiac fibrosis is a hallmark of cardiovascular dysfunction, where ventricular wall thickening is a consequence of excessive extracellular matrix deposition, which can reduce cardiac contractility [13]. Literature suggests that LOXL2 expression in the heart needs to be tightly regulated in order to prevent myocardial fibrosis, with these articles summarized in Table 1. Briefly, a study confirmed a positive correlation between increased LOXL2 mRNA expression and the development of cardiac dysfunction, in both transgenic mice and humans [24]. In human studies performed by Raghu et al. and Zhao et al. (2017), Raghu et al. showed that admission of Simtuzumab, which binds to LOXL2 in the treatment of fibrosis, did not improve the health outcomes of the patients with idiopathic pulmonary fibrosis [74]; however, Zhao et al. showed that patients with atrial fibrillation had increased serum LOXL2 levels, which was associated with increased left atrial size; however, there was no effect on left ventricular function [25]. In the same year, Mižíková et al. (2017) found that treating both lung fibroblasts and C57Bl/6J mice with a general inhibitor of the LOX gene family had no effect on the mRNA expression of these genes using both an in vivo and in vitro model [75]. Additionally, Craighead et al. (2018) treated hypertensive patients with the same LOX inhibitor, and found an increase in ECM-bound LOXL2 expression in these patients [76]. By means of proteomic analysis, Steppan et al. (2018) confirmed that LOXL2 mediates the stiffening of smooth muscles cells in aging using a LOXL2 knock-out mouse model [77]. This was confirmed by Torregrosa-Carrión et al. (2019), who reported that NOTCH activation led to increased expression of TGF-β2 and collagen, which form part of the LOXL2 signaling pathway [78]. Lastly, it was shown by Schilter et al. (2019) that administration of a LOXL2 inhibitor for 4-weeks, after left coronary arteries occlusion, resulted in an observed decreased myocardial fibrosis with improved cardiac output in a C57/BL6 mouse model [79]. Although there is data linking LOXL2 to fibrosis, its regulation in the fibrotic cardiac disease pathophysiology remains ill-defined, and as such, more research is required to better define LOXL2′s mechanistic role in cardiac tissue fibrosis and subsequent contractile dysfunction. 

## 5. LOXL2 Activity and Its Gene Regulatory Network

Several mechanisms of LOXL2 regulation have been proposed (Figure 4). Interestingly, galectin-3 (GAL3), although not directly related to LOXL2, is also involved in the fibrotic process through its interaction with TGF- β. Although both proinflammatory and anti-inflammatory effects have been suggested for GAL3 [80,81], it has been mostly implicated in the development and progression of HF [82,83]. LOXL2 forms part of the PI3K/AKT/mTOR pathway [24], which is able to upregulate the hypoxia-inducible factor 1 (HIF-1). This protein plays a critical role in cardiac oxygen homeostasis, and its dysregulation results in ischemic heart disease [84,85]. Nevertheless, the PI3K/AKT/mTOR pathway can activate TGF-β signaling, thereby prompting enhanced activity of the fibrotic pathways [24]. A review by Cox and Erler suggested that tissue fibrosis is caused by specific stressors, which activates TGF-β, cytokines and other inflammatory responses [86]. This in turn could result in the recruitment and subsequent activation of fibroblasts to myofibroblasts, which are then able to produce α-SMA and upregulate proteins such as connective tissue growth factor (CTGF), LOX and LOXL2 [86]. This leads to the promotion of collagen deposition and cross-linking, resulting in the modification of the ECM, tissue stiffening and organ failure [86], making LOXL2 and its gene regulatory network an ideal drug target to protect against cardiac fibrosis and improve heart function.

In addition, Chaudhury reported that signal transducers for TGF-β receptors, SMADs, are important regulators of cellular growth and development through their interaction with TGF-β [87]. Research done by Min Lu et al. (2019) on the regulation of LOX expression suggested that TGF-β1 plays a pertinent role in the phosphorylation and subsequent activation of SMAD2 and SMAD3 signaling [88], with these molecules subsequently being translocated into the nucleus where they interact with DNA-binding factors such as activating protein 1 (AP-1), specificity protein 1 (Sp-1) and NFκB [89] to have downstream effects on gene expression. *LOXL2* is known to have binding sites for all three the aforementioned DNA-binding factors and thus its expression can be induced through TGF-β1/SMAD2/3 signaling [28]. Interestingly, the SMAD proteins have been found to be present in CVDs or after an ischemic cardiovascular event, where they are involved in the initiation of the fibrotic processes [89]. Moreover, when TGF is inhibited, a decrease in cardiac interstitial fibrosis as well as a suppression of late stage cardiac remodeling was observed after a myocardial infarction in mice [90]. There is also evidence in a cardiac reperfusion injury model that inhibition of the renin-angiotensin aldosterone system (RAAS) is associated with reduced levels of TGF-h and SMAD activity, together with a decrease in fibrosis [91]. Lu et al. (2019) also showed a decrease in c-Jun, one of the components of the AP-1 DNA-binding factors, caused a subsequent decrease in LOX expression as well as downstream effectors such as collagen [88]. Taken together, this suggests that TGF-β may act via SMAD and AP-1 to regulate LOX expression, increasing cardiac fibrosis. Although this proposed pathway has not yet been evaluated for LOXL2 by any single research group, it is plausible to assume, that based on the similar structure to LOX, with respect to the regulatory sites for DNA-binding factors, that LOXL2 is regulated in a similar manner to LOX. Nonetheless, in the search to better understand the role that LOXL2 plays in CVD, it is also important to understand the involvement of epigenetic modification.

## 6. Epigenetic Control of LOXL2 Expression

Epigenetic modifications are defined as heritable changes in chromatin structure and DNA expression, with no alteration in the DNA sequence [92]. The major epigenetic modifications are DNA methylation (the addition of methyl groups to cytosine residues of CpG sites), acetylation (the addition of acetyl groups to lysine residues) and histone modification (methylation or acetylation of histones, altering the chromatin structure). These all influence the regulation of gene expression. MicroRNAs (miRNAs) are also a form of epigenetic modification that plays a role in regulating gene expression. Interestingly; however, miRNA expression has also been shown to be controlled by epigenetic modifications [93,94]. Although epigenetics have been implicated in controlling LOXL2 expression, a paucity of data still exists regarding whether these modifications are key in its regulation.

The *LOXL2* gene is located on chromosome 8p21.3 [95,96] and contains 5 CpG sites within the promotor region and the first exon; these known sites could be influenced by DNA methylation [28,97]. LOXL2 expression is also believed to be controlled through histone modification. Fong et al. (2007) [28] and a later study done by Hollosi et al. (2009) [97] showed that treatment with a histone deacetylase inhibitor resulted in a change in LOXL2 expression, indicating that histone modification influenced LOXL2 expression. Although miRNAs in LOXL2 regulation have not been studied in CVDs, evidence from cancer studies have shown that miRNAs can regulate LOXL2 expression in tumors [98]. Since epigenetic modifications have been linked to the development and progression of various diseases, understanding how these mechanisms regulate LOXL2 could provide new avenues to control its expression, and potentially be advantageous in the treatment of fibrosis induced CVDs.

### 6.1. LOXL2 and DNA Methylation

DNA methylation can be affected by environmental factors [99], and is a reversible process, making it an attractive therapeutic approach. The methylation reaction is catalyzed by DNA methyltransferases (DNMTs) [100], but its dysregulation in cellular processes by both hyper- and hypomethylation contributes to the development and progression of diseases [100].

Methylation sites within a genes promotor region are important in the regulation of gene expression [101]. Fong et al. [28] and Zhong et al. [43] both showed that methylation within the *LOXL2* promotor region aids in the regulation of its expression. In addition to this, an in vitro study done on human chondrocytes showed an increase in inflammatory markers resulted in a decrease in LOXL2 expression [102]. Dong et al. (2017) also argued that an inverse relationship exists between DNA methylation and inflammation in healthy African-American teenagers [103]. It is also known that transcription factors, such as NF-κB, and changes in DNA methylation status as well as histone modifications are responsible for the regulation of genes associated with inflammatory pathways [104]. Inflammatory factors are known to upregulate the expression of LOXL2 and thereby increase fibrosis [105]. In addition, an epigenome-wide association study conducted on Asian and European cohorts showed that genes that are known to have an association with hypertension were differentially methylated, and that there were differences between the methylation statuses of the two cohorts [106]. It was also shown in a mouse model that if DNMT1, an enzyme that catalyzes methylation, is silenced, an upregulation is seen in the interleukins and cytokines, thus increasing inflammation [107]. Biological processes of inflammation and high blood pressure, which are both risk factors for the development of CVDs [43], can be controlled by changes in methylation, thus making the discovery of methylation targets important in combating fibrotic diseases.

### 6.2. LOXL2 and Histone Modification

In addition to methylation, it has been shown that histone modification also plays a role in LOXL2 regulation [28]. Histones are protein structures within chromosomes around which the DNA strands or chromatin are coiled. Histone modification aids in the packaging of DNA with thus influencing gene expression [108]. Although there is little evidence to link histone modification and LOXL2 expression, evaluations in breast cancer cell lines reveal that treatment with a general DNA methylation inhibitor, 5-azacytidine, resulted in a 40-fold upregulation of *LOXL2* mRNA expression, and further co-treatment with a histone deacetylase inhibitor, trichostatin A, resulted in a further 5-fold expression increase [28,97]. It has; however, been shown that an increase in histone acetyltransferase post-myocardial infarction has a positive impact on myocardial remodeling, specifically within the left ventricle [109], and thus, histone modifications could play a key role in the pathogenesis of cardiac fibrosis. More research, however, needs to be conducted surrounding LOXL2 and regulation of its expression by histone modifications.

## 7. Future Considerations

The majority of LOXL2 mechanistic studies have been focused on cancer. To date, it is known that the expression of LOXL2 is tightly regulated, with augmented LOXL2 expression being linked to interstitial fibrosis. Nonetheless, to date, there is a paucity of data linking LOXL2 mechanistically to cardiovascular dysfunction. As such, future studies should include functional studies that attempt to inhibit LOXL2 to better understand it role in the fibrotic process. Although there is no treatment for fibrosis in the myocardium, recently two drugs, pirfenidone and nintedanib have been approved for the treatment of idiopathic pulmonary fibrosis and exert their effects by inhibiting TGF-β, platelet-derived growth factor (PDGF) and FGF receptors, respectively, which are known regulators of LOXL2 expression [110]. What is still unknown however, is how LOXL2, through the ECM, causes changes in TGF-β signaling, and whether LOXL2 has a direct effect on the myofibroblasts to result in their migration. Deeper investigation into the signaling mechanisms driving these processes could be key in identifying other important effectors in the fibrotic process.

Additionally, although some human studies have been conducted [24,25,74,76], characterization of LOXL2 expression and function within the heart is limited. Experimental data need to be validated in humans, in both male and female patients, and non-invasive methods will need to be developed to test these markers by correlations with other risk factors in order to be used as a possible predictor of CVD. A further limitation to current research is that the role of epigenetic effects influencing LOXL2 expression has not been evaluated. Although DNA methylation and histone modification have been implicated in the regulation of LOXL2 expression, it is known that there is an interplay between miRNAs and epigenetic mechanisms [111,112], and thus the effect of miRNAs on LOXL2 gene expression should also be explored.

Epigenetic changes can be regulated by external factors such as smoking, pollution, diet and exercise [113]. Diet, such as the Mediterranean diet, characterized by high polyphenol content is also associated with reduced cardiovascular risk, and as such, should be investigated [114]. Interestingly, polyphenols are also known to have the potential to modulate epigenetic machinery; however, the mechanisms by which polyphenols exert their effects, particularly in affecting *LOXL2* methylation or histone modifications, are yet to be elucidated. Additionally, it is known that polyphenols act as anti-inflammatory agents, reducing the stressors that could activate the fibrotic pathways, and could thus prove to be useful in this regard and warrants further investigation.

## 8. Conclusions

This review presents evidence that, under conditions of stress, increased inflammation activates NFκB, stimulating LOXL2 expression, as well as other factors like TGF-β, COL1A and CTGF, which in turn, lead to elevated collagen deposition and cross-linking, culminating in increased fibrosis. There is, however, a paucity of available literature, thus hindering efforts to fully elucidate not only the exact mechanisms by which LOXL2 signaling occurs in fibrosis, but also the role that epigenetics plays in LOXL2 and cardiovascular disease progression. There is a definite need for further investigating the effects of epigenetics in LOXL2 expression, which could greatly aid in identifying a means to control and manipulate LOXL2, particularly in humans, thereby developing novel therapeutics to combat this worldwide CVD problem.

## Figures and Tables

**Figure 1 ijms-21-05913-f001:**
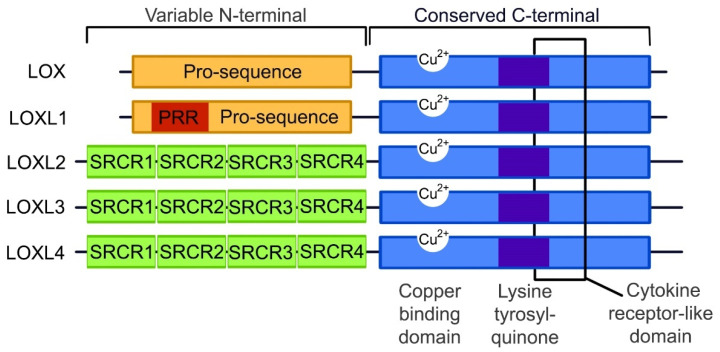
The structure of the LOX and LOXL1–4 proteins. The C-terminal is conserved between all the LOX protein family members, containing a copper binding domain, a lysine tyrosylquinone cofactor residue and a cytokine receptor domain. At the N-terminals, LOX and LOXL1 contain pro-sequences, with LOXL1 containing a proline-rich region (PRR), while LOXL2–4 contain scavenger receptor cysteine-rich domains (SRCR) within the *N*-terminal. (Image adapted from Wu 2015 [50]).

**Figure 2 ijms-21-05913-f002:**
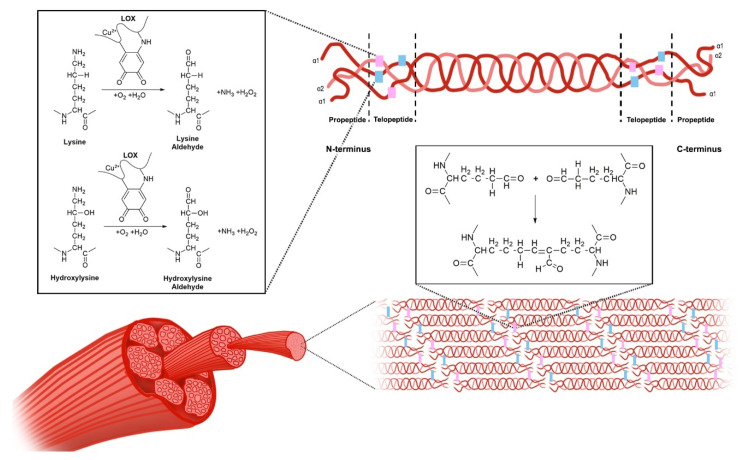
The mechanism of lyslyl oxidase collagen cross-linking. LOX catalyzes the conversion of lysine and hydroxylysine to lysine aldehyde and hydroxylysine aldehyde, respectively. This occurs within the telopeptide region of the procollagen molecules. The propeptide fragments of these molecules are then truncated to form tropocollagen molecules, which self-assemble and form cross-links, thereby forming collagen fibrils.

**Figure 3 ijms-21-05913-f003:**
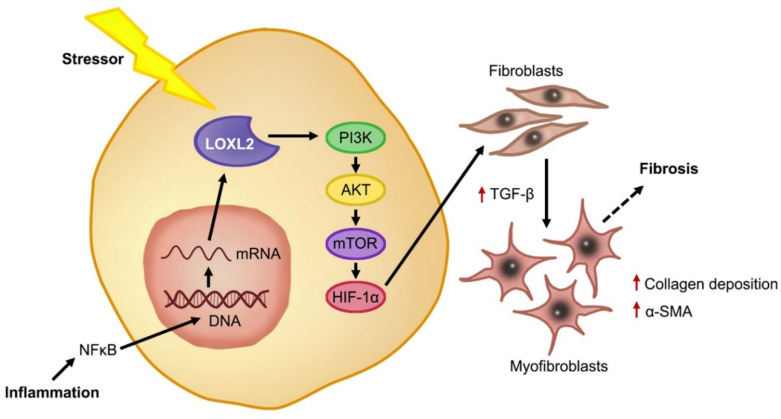
LOXL2-induced cardiac fibrosis. Under stress conditions such as inflammation, the activation of NFκB causes increased mRNA expression of *LOXL2* and downstream, LOXL2 activates the PI3K/AKT/mTOR pathway, increasing TGF-β and triggering fibroblasts differentiation, where myofibroblasts secrete α-SMA and increase collagen deposition. Overstimulation of this process results in ECM deposition and fibrosis.

**Figure 4 ijms-21-05913-f004:**
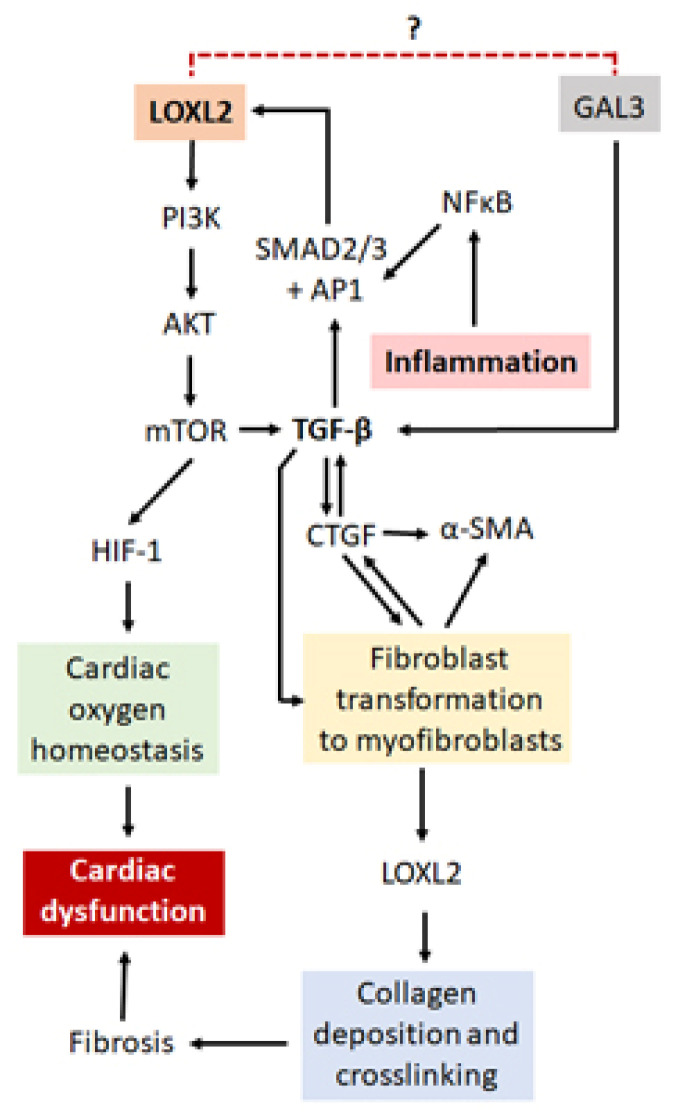
A proposed mechanism by which LOXL2 causes fibrosis and cardiac dysfunction. LOXL2 acts via the PI3K/AKT/mTOR pathway to activate TGF-β signaling and HIF-1 protein expression. GAL3 also activates TGF-β signaling, thus with possible similar effects as LOXL2. TGF-β signaling results in an increase in α-SMA, CTGF and LOXL2 expression, which leads to an increase in collagen deposition and cross-linking, resulting in fibrosis, ventricular stiffness and cardiac dysfunction. Inflammation causes an increase in NFκB which interacts with AP1 and Sp-1 proteins, also increasing LOXL2 expression. Dysregulation of HIF-1 protein expression results in the disruption of oxygen homeostasis in the heart, also having pathological effects. Further investigation is needed to find out whether there is a direct interaction between LOXL2 and GAL3.

**Table 1 ijms-21-05913-t001:** Summary of articles investigating LOXL2 and fibrosis in cardiovascular disease.

Species	Study Design	Findings	References
Loxl2+/− knockout mice	Mice: Underwent transaortic constriction followed by LOXL2 expression analysis and histology.	Transgenic mice: cardiac stress results in↑ LOXL2 → myocardial fibrosis & dysfunction. Inhibition of LOXL2 activity: ↓ cardiac fibrosis and ↑ cardiac function.	Yang et al. (2016) [24]
Human	Human: Patients presenting with HFpEF and diastolic dysfunction without symptoms underwent right-ventricular biopsies for evaluation of cardiomyopathy.	LOXL2 acts via the PI3K/AKT pathway to activate TGF-β2. Diseased human hearts: LOXL2 ↑ in the interstitial space and serum ↑ LOXL2 expression correlated with ↑ fibrosis and myocardial dysfunction.	
Human	Patients (aged 45–85) with idiopathic pulmonary fibrosis were treated with simtuzumab or a placebo once a week and its effects studied.	Simtuzumab, did not improve survival rates in patients with idiopathic pulmonary fibrosis.	Raghu et al. (2017) [74]
Human	Patients with atrial fibrillation were assessed in terms of serum LOXL2 levels, left atrial size and left ventricular function.	Atrial fibrillation patients: ↑ serum LOXL2 Positively associated with increased left atrial size.	Zhao et al. (2017) [25]
Primary cells isolated from C57Bl/6J mice, macrophages and endothelial cells, and mouse pups	Primary cells: cultured in the presence of a LOX inhibitor or *LOX*, *LOXL1* and *LOXL2* knocked down with siRNA. Gene expression, amine oxidase activity and microarray analyses were performed Mice: a bronchopulmonary dysplasia model was established, and lungs harvested for expression analysis.	*Lox*, *Loxl1*, and *Loxl2* are highly expressed in primary mouse lung fibroblasts. Knockdown of *Lox*, *Loxl1*, and *Loxl2*: associated with change in gene expression (primary mouse lung fibroblasts). BAPN: no impact on mRNA levels of LOX target-genes, in lung fibroblasts or in BAPN-treated mice.	Mižíková et al. (2017) [75]
Human	Intradermal microdialysis fibers were placed in the forearm of young, normotensive and hypertensive individuals. Fibers treated with β-aminopropionitrile, a LOX inhibitor, or acted as a control. Norepinephrine was used to examine the vasoconstrictor function and sodium nitroprusside to study smooth muscle vasodilation.	LOX inhibition augmented vasoconstrictor sensitivity in young and normotensive but not hypertensive patients. ECM-bound LOX expression: ↑ in hypertensive subjects vs. younger patients. Vascular stiffness & microvascular dysfunction in hypertension could be due to ↑ LOX expression.	Craighead et al. (2018) [76]
Human aortic smooth muscle cells and LOXL2+/− mice	Human aortic smooth muscle cells were cultured and the secretome analyzed. Mice: nitric oxide production was assessed in the aortic rings.	Proteomic analysis: LOXL2: important mediator of age-associated vascular stiffening in smooth muscle cells. Nitric oxide assessment: it ↓ LOXL2 abundance and activity in the ECM of isolated smooth muscle cells. Knock out mice: protected from age-associated vascular stiffening. Isolated aortic rings: LOXL2 mediates vascular stiffening in aging by promoting smooth muscle cell stiffness, contractility, and matrix deposition.	Steppan et al. (2018) [77]
Mouse embryonic endocardial cells, human aortic smooth muscle cells and LOXL2+/− mice	Mouse embryonic endocardinal cells were stimulated with DLL4 and JAG1, with or without NOTCH inhibitors. Proteomics analysis of the media was conducted to identified proteins that are secreted in response to NOTCH signaling manipulation.	Secretome analysis identified 129 factors that showed a change in expression when NOTCH was activated or repressed. NOTCH activation correlated with ↑ expression of TGF-β2 and collagen.	Torregrosa-Carrión et al. (2019) [78]
Wistar rats, Sprague Dawley rats, C57/BL6 mice	A LOXL2/LOXL3 inhibitor, PXS-5153A, was developed and its effect on LOXL2/3 in relation to collagen cross-linking and fibrosis was assessed.	PXS-5153A ↓ collagen cross-linking in vitro. PXS-5153A ↓ collagen expression and cross-linking, thereby ↑ liver function. In a model of myocardial infarction, addition PXS-5153A, ↑ cardiac output. This shows that inhibition of LOXL2/LOXL3 activity could be a viable treatment option for liver fibrosis.	Schilter et al. (2019) [79]

Symbols: → = leads to; ↓ = decreases; ↑ = increases.

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
