# Peer review of "Linking LOXL2 to Cardiac Interstitial Fibrosis"

_ijms, 2020, doi:10.3390/ijms21165913_

Round 1

Reviewer 1 Report

The manuscript by Erasmus and co-authors describes the role of LOXL2 cardiac fibrosis, with an emphasis in understanding its molecular mechanisms of action. In addition, modulation of LOXL2 expression by DNA methylation and histone modification is evaluated. While this is an interesting topic, there is however an overall paucity of data on LOXL2 function and regulation in cardiac tissue.

Major comments:

1) Given that there is currently limited experimental evidence regarding the function of LOXL2 in the heart, it seems unclear as to why the authors chose to prepare a review article on its role in cardiac fibrosis. 

2) Since LOX and the LOXL1-4 isoforms have conserved catalytic domains and appear to have similar functions, it is unclear why the authors chose to discuss specifically LOXL2 in this manuscript. And this is in particular since there is limited experimental evidence on its role in the heart.

3) As mentioned in the abstract of the manuscript, lines 27-28 ‘this review aims to elaborate on the current understanding of the mechanisms by which LOXL2 is regulated in physiological conditions’. However, at later points the author clarify that there is limited knowledge on the functional significance of LOXL2 in the physiology of the heart but instead most information available is based on disease models. Therefore the aim of the review is not achieved.

4) Given that the title of the review refers to cardiac fibrosis, the manuscript should be more focused on the role of LOXL2 in cardiac tissue. Therefore, parts referring to non-muscle findings of LOX (eg paragraph on page 5 referring primarily to bone studies) should be removed.

5) Repetition of introductory information should be limited (eg page 7, lines 230-233 describing diabetes, or lines 242-245 that are describing again cardiac fibrosis).

6) In the introduction, a brief description of collagen cross-linking mechanism should be included

7) Section 6.2 should be removed since there is still very limited evidence regarding the role of LOXL2 and histone modification. Moreover, given the lack of experimental evidence, in its current form this section is mainly background information regarding histone modification.

Minor comments:

1) Page2, Line 88 ‘plays a crucial role in cellular structure and repair, as well as’, there is a word missing after as well as

2) Page2, Line 94, ‘forcesblood’, please correct

Author Response

Thank you for taking the time to review our manuscript. Please see the attachment for the response to comments.

Reviewer 2 Report

I enjoyed reading your comprehensive, well-written and informative review. Please consider the following minor suggestions:

It would be helpful to have early paragraph describing the structure of Type-1 Collagen, distinguishing it from other collagen types and briefly explaining the chemical nature of 'cross-links'

L 94 Typo "forcesblood"

Author Response

(The authors gave the same response as above.)

Round 2

Reviewer 1 Report

Thank you for addressing my comments and concerns. I believe that this has helped towards improving the manuscript, which can now be accepted for publication.